# Association between Birth Weight and Subcutaneous Fat Thickness at Adulthood in Dogs

**DOI:** 10.3390/vetsci10030208

**Published:** 2023-03-09

**Authors:** Amélie Mugnier, Fanny Cellard, Anthony Morin, Magalie Delmas, Aurélien Grellet, Sylvie Chastant

**Affiliations:** 1NeoCare, Université de Toulouse, ENVT, 31076 Toulouse, France; 2CESECAH, Lieu-dit Monsable, 63190 Lezoux, France; 3Ecole des Chiens Guides d’Aveugles du Grand Sud-Ouest, 31500 Toulouse, France

**Keywords:** dog, body condition score, ultrasound, subcutaneous fat thickness, birth weight, Labrador

## Abstract

**Simple Summary:**

The growing body of evidence for a Developmental Origin of Health and Diseases (DOHaD) emphasizes the need to assess the role of fetal and neonatal factors in canine overweight which affects nearly 40% of adult dogs. The purpose of the current study was to examine the association between birth weight and subcutaneous fat thickness (SFT) at adulthood, measured by ultrasonography, in a population of purebred Labrador dogs. We used a linear mixed-effects model by adjusting for sex, age, neuter status and the anatomical sites of SFT measurements (abdomen, flanks and lumbar region). The results suggest that, as in other species, dogs with the lowest birth weights have a greater thickness of subcutaneous fat at adulthood than the others. These findings highlight the importance of fetal life in adult health in the canine species. They suggest that birth weight is an important health indicator of long-term impact. It should be considered in particular in overweight management plans: puppies born with low birth weight should be identified as being at higher risk of overweight and their owners should be encouraged to enter their dogs in specific weight control plans from an early stage in life.

**Abstract:**

Overweight affects nearly 40% of dogs. The objective of this study was to explore the hypothesis of the Developmental Origins of Health and Disease through the association between birth weight and adiposity in adult dogs. The association between body condition score (BCS) and subcutaneous fat thickness (SFT), measured in the flanks, abdomen and lumbar regions, was assessed in a population 88 adult Labradors (>1 year). Significant positive moderate correlations between BCS and SFT were described. A linear mixed-effects model was used to investigate the association between birth weight and SFT by adjusting for sex, age, neuter status and the anatomical site of the measurement. The results showed that SFT values increased with age and were higher in sterilized than in entire dogs. In addition, SFT values were higher in the lumbar region compared to the other anatomical sites. Finally, the model revealed a significant association between SFT and birth weight, suggesting that, as in other species, dogs with the lowest birth weights have thicker subcutaneous fat at adulthood than the others. The assessment of visceral adipose tissue and the relative importance of birth weight among the numerous risk factors of overweight remains to be explored in dogs.

## 1. Introduction

In the canine species, overweight affects nearly 40% of dogs [1,2,3,4,5] and is becoming a major population health issue [6]. Much more than a simple aesthetic slight, overweight and, a fortiori, obesity, predispose animals to numerous pathologies (osteoarthritis, cardiopathies, diabetes, constipation, dermatitis, anesthetic risk, etc.) [7,8], which contribute towards reducing their life expectancy [9,10], particularly their healthy life expectancy and, more globally, their comfort of life [11,12]. As excess adiposity is known to be difficult to reverse, it appears critical to prevent it as early as possible in life.

Since the 1980s and the development the “Barker hypothesis”, suggesting an impact of early life events on the risk of diseases in adulthood, a growing body of epidemiological and experimental data have supported the Developmental Origins of Health and Disease (DOHaD) in humans [13,14,15,16]. Events undergone during early life including embryonic and fetal phases have long-term consequences until adulthood and elderly ages. An indicator of a suboptimal in utero environment, low birth weight was demonstrated to be associated with excess adiposity at adulthood in many mammalian species such as humans [17,18,19], pigs [20], mice [21] and guinea pigs [22]. In the canine species, maternal programming is just beginning to be explored. A previous work, on a Labrador dog cohort, demonstrated an association between low birth weight and increased risk of overweight at adulthood, assessed through the body condition score (BCS) [23]. This score, established by palpation and visual observation on a 9-point scale, is widely used for body composition assessment both in clinical practice and for research because it is inexpensive, non-invasive and provides immediate results [24,25,26]. Although its correlation with dual-energy X-ray absorptiometry and deuterium oxide dilution is well established [25,27], studies have shown its subjectivity [28,29] with the influence of scorer experience [30,31]. In order to give a more precise estimation of canine body composition and more particularly adiposity, more reliable and objective descriptors have been developed such as the measurement of subcutaneous fat thickness (SFT). SFT, measured by ultrasonography at several anatomical sites is correlated with the proportion of body adipose tissue in canine species [32].

The objective of this study was to explore further the association between birth weight and adiposity at adulthood, assessed through the SFT, in a Labrador dog population.

## 2. Materials and Methods

### 2.1. Study Population

#### 2.1.1. Dog Population

The study population was 88 adult (>1 year), purebred Labradors, born in one breeding kennel (Centre d’Etude de Sélection et d’Elevage pour Chiens Guides d’Aveugles et Autres Handicapés, CESECAH; Lezoux, France). At the time of examination, they were either active, retired or breeding assistance dogs and they are housed in volunteer families. Age, sex and neuter status were recorded and weights at birth were extracted from the kennel database.

#### 2.1.2. Assessment of Adiposity

For each dog, adiposity was estimated through two different methods. First, the body condition score (BCS) was assessed jointly by two operators, through palpation and visual observation, using the 9-point scale developed by Laflamme [25], i.e., from 1 (emaciated) to 9 (grossly obese). Any disagreement between the two operators was resolved by consensus. Subcutaneous fat thickness (SFT) was measured by real-time ultrasonography (MyLab One, ESAOTE, Hospimedi, Pouilly, France and variable frequency microconvex probe 6–13 MHz) at three locations (abdomen, flanks and lumbar region; Figure 1) using the method previously described by Payan-Carreira et al. [32]. For evaluation on the abdomen, the probe was placed in vertical position on the left lateral wall of the abdomen, midway between the linea alba and the tip of the transverse process of the lumbar vertebra. For evaluation on flanks, the probe was placed transversely to the rib on the external oblique abdominal muscle on the left side of the dog, over the 9th intercostal space and just above the costochondral junction. For evaluation at the lumbar region, the probe was positioned between the 3rd and 5th lumbar vertebrae, parallel to the spinous process, 2 to 3 cm to the left of the midline. Dogs were in standing position, no anesthesia, sedation or clipping was required and ethanol served as a coupling medium. For each location, three measurements were taken and the average was then used for the analysis.

### 2.2. Statistical Analysis

The information collected was transferred into an Excel spreadsheet (Microsoft Corporation, Redmond, Washington, DC, USA). Then, statistical analysis and graphical representations were performed using R software, version 4.2.1 [33] and the following packages: corrplot [34], emmeans [35], FSA [36], ggplot2 [37], lme4 [38], performance [39] and rcompanion [40]. Statistical significance was defined as *p* value < 0.05 and statistical uncertainty was assessed by calculating the 95% confidence intervals (95% CI).

After a description of the population studied (Appendix A), the relationship between BCS and SFT for each point of measurement was tested with the Kruskal–Wallis rank sum test followed by a Dunn test with Benjamini–Hochberg correction.

Then, the interplay between birth weight and SFT values was investigated by fitting a linear mixed regression model with SFT as response variable. Explanatory variables included birth weight (categorized at LTM, Lower Than the Median/HTM, Higher Than or equal to the Median), the animal’s sex (male/female), age (young, less than 2 years included/adult, 2–5 years/old, more than 5 years included), neuter status (neutered/entire) and location (abdomen/flanks/lumbar region). Dog identification was introduced as a random term to deal with the repetition of the individuals (three locations per dog). Starting with the full model (SFT~Location + Birth weight + Neuter status + Sex + Age), a backward selection based on Akaike’s Information Criterion (AIC) was applied to select the most parsimonious model. At the end of the backward selection, interactions between birth weight and location as well as between birth weight and age were tested by running models with and without the interaction term. The normal probability plot of residuals and the plot of residuals versus predicted values were generated to check whether the assumptions of normality and homogeneity of variance had been fulfilled. The estimated marginal means of SFT were obtained from this model and pairwise comparisons were performed using Tukey’s Honest Significant Difference method. 

## 3. Results

### 3.1. Population

Among the 88 adult Labradors included in the present study, ages ranged from 1.1 to 9.6 years (median: 3.4 years) with 28 young dogs (≤2 years), 29 adults (2–5 years) and 31 old dogs (≥5 years). Sex distribution was 6 (6.8%) entire males, 34 (38.6%) entire females, 17 (19.3%) neutered males and 31 (35.2%) spayed females. BCS varied between 4 and 9 (*n* = 9, *n* = 27, *n* = 25, *n* = 17, *n* = 8 and *n* = 2, for BCS levels 4, 5, 6, 7, 8 and 9, respectively; Table 1) and the overall prevalence of overweight (BCS ≥ 6) was 59.1% (95% confidence interval, 95% CI: 48.1–69.5). The median birth weight was 410 g (interquartile range, IQR: 380–470) with 43 dogs in the HTM group (birth weight ≥ 410 g) and 45 dogs in the LTM group (birth weight < 410 g).

### 3.2. Association between SFT and BCS

SFT, assessed at three different locations (abdomen, flanks and lumbar region), varied between 0.8 and 35.6 mm (median: 7.9 mm; Table 1). The correlation matrix built between the three measurement sites and BCS described positive significant but moderate correlations (Figure 2). The highest correlation was obtained between SFT from the flanks and BCS (r = 0.69; *p* < 0.001). For each site, SFT significantly increased with BCS (*p* < 0.001 for all, Kruskal–Wallis rank sum test; Figure 3).

### 3.3. Association between Birth Weight and SFT

The results of the final multivariable model of the association between SFT and birth weight categories are summarized in Table 2. The fixed effects explained 40% of the variation in the SFT observed in the study population and the full model (i.e., including random term) explained 56% of this variation.

The estimated marginal mean of SFT was higher in old Labradors compared with the two other groups (11.29 mm, 95CI: 10.28–12.30 vs. 7.77 mm, 95CI: 6.73–8.81 for adult dogs and 8.23 mm, 95CI: 7.20–9.26 for young dogs). It was also higher for sterilized dogs (9.72 mm, 95CI: 8.92–10.52) than for entire dogs (8.47 mm, 95CI: 7.61–9.34) and in the lumbar region compared with the two other locations (12.52 mm, 95CI: 11.72–13.32 vs. 7.08 mm, 95CI: 6.28–7.88 for flanks and 7.69 mm, 95CI: 6.90–8.49 for abdomen). Finally, the estimated marginal mean of SFT was higher for dogs with birth weights below the median compared to those with higher birth weights (9.85 mm, 95CI: 9.02–10.7 vs. 8.35 mm, 95CI: 7.50–9.2). Individuals with the lightest birth weights had thicker fat deposits whatever the site of measurement (abdomen, flanks, lumbar region; Figure 4a) and at all periods of adulthood (Figure 4b). None of the interactions tested (birth weight and location or birth weight and age) were significant.

## 4. Discussion

In contrast to the differences with mouse, rabbit and human, evidence of maternal programming in the canine species is lacking [41]. This work provides the demonstration of an association between birth weight (endpoint of embryofetal growth) and adiposity at adulthood in dogs. Based on a quantitative assessment of adiposity, i.e., subcutaneous fat thickness measured by ultrasonography, the present study strengthens results from a previous study observing a link between birth weight and BCS [23]. The population studied was chosen in one breed predisposed to overweight (the Labrador) with all dogs born in the same kennel and raised under similar environmental conditions until two months of age.

Although the measurement of SFT requires equipment (a real-time ultrasound machine) contrary to BCS, its use in livestock (pigs [42,43], small ruminants [44,45], horses [30] and cows [28,46,47]) suggests the feasibility of its implementation in canine practice. Indeed, SFT ultrasound measurement does not require sedation and is a quick, non-invasive and easy to learn objective examination [28,32]. In our study, the SFTs measured ranged from 0.8 to 35.6 mm and were higher than the measurements obtained by Payan-Carreira et al. (1 to 7.9 mm) [32] or by Wilkinson et al. (0.4 to 4.7 mm) [48] for the three locations, and by Morooka et al. for the lumbar region (0.8 to 9.1 mm) [49]. This difference could be explained by the absence of grossly obese dogs in the two studies cited previously (BCS ranging from 1 to 4 on a 5-point scale in Payan-Carreira et al. and from 4 to 8 on a 9-point scale in Morooka et al.) [32,49]. In addition, differences in body composition depending on the breed have been suggested [46] and the Labrador is considered as predisposed to become overweight compared to other canine breeds [47,48] which could lead to higher adiposity and thus higher SFT values. 

BCS is a semi-quantitative subjective parameter for body composition whereas SFT is more quantitative and objective. Nevertheless, both criteria were correlated in the present work, as in previous studies [32,48,49]. Among the three locations considered, SFT measurements from the flanks were the most positively correlated with BCS (r = 0.69) but only slight SFT differences between the various BCS scores were observed (Figure 3). This higher correlation between BCS and SFT at the flanks suggests that BCS attribution is strongly influenced by the palpation/observation of fat deposits on this area. SFT at the lumbar region was also well correlated with BCS (r = 0.62) with a gradual increase, as shown in Figure 2, suggesting that this anatomical area may be useful to assess dog body composition in dogs, such as back fat thickness in livestock species [43]. As in humans, fat deposition varied depending on the anatomical region [50,51], subcutaneous fat being thicker in the lumbar area compared to the flanks and abdomen. Among variation factors, as reported elsewhere in dogs [52,53] and in other species (humans [54] and lambs [55]), the adiposity of adult dogs increased with age and neuter status. Contrary to what has been observed in humans [50,56,57], lambs [55] or pigs [58], no influence of sex on SFT values was observed in our population of Labradors. Further studies, including breed effect, are warranted to explore the parameters involved in the pattern of subcutaneous fat deposition in the canine species.

The main objective of this paper was to assess whether birth weight belongs to the variation factors of SFT at adulthood: adult dogs with the lightest weights at birth exhibited higher values of SFT at all locations and at all ages. This finding is in agreement with the literature indicating the maternal programming of adiposity at adulthood by low birth weight [59,60,61]. Given the sample size and to be consistent with our previous study [23], the effect of birth weight was explored by separating the dogs into two groups relative to the median birth weight in the population. To further explore and refine this relationship between birth weight and adult adiposity, it may be of interest to study the effect of more extreme values of birth weight (e.g., first and fourth quartiles or first and tenth deciles).

As the interaction between birth weight and location was not significant, birth weight can be considered as a determinant of later subcutaneous fat deposition but without any impact on its distribution pattern. Our results are in contradiction with previous work on humans indicating a more truncal distribution of subcutaneous fat in adults with low birth weight [61]. This might be explained by major differences in body fat distribution, as highlighted by Kempster [62]. Nevertheless, both BCS and SFT are used to assess only peripheral fat deposition whereas recent findings in humans suggest the importance of the distribution of adipose tissue, subcutaneous or visceral, regarding the risk of overweight co-morbidities [63,64]. For example, visceral adiposity has been described to be a better predictor than SFT of incident impaired glucose tolerance, one of the major co-morbidities associated with obesity in humans [65,66]. In humans and lambs, low birth weight was found associated with visceral adiposity [67,68,69]. Accurate methods of body composition assessment such as dual-energy X-ray absorptiometry [70] would allow exploring the impact of birth weight on body fat distribution, including visceral adiposity, at adulthood in the canine species.

These findings highlight the importance of fetal life in adult health in the canine species. They suggest that birth weight is an important health criterion of long-term impact. It should be considered in particular in overweight management plans: puppies born with low birth weight should be identified as having a higher risk of overweight and their owners should be encouraged to involve the dog in a specific weight control plan early in life. In addition, it would be wise to consider training pet owners who currently tend to underestimate the adiposity of their animal [31,71]. This would allow them to detect overweight as early as possible, without waiting for a visit to the veterinarian.

Maternal programming in the canine species is therefore a public health issue that requires the involvement of all stakeholders: breeders are responsible for newborns and for passing on the value of birth weight to owners; veterinarians need to pay attention to birth weight as a significant health outcome even in adulthood; and owners need to be aware of this risk factor for overweight and seriously follow a weight management plan. In addition to birth weight, further studies are required to assess the influence of neonatal events and namely early growth on overweight risk at adulthood.

Many other factors suggested or demonstrated to have effects on adiposity in dogs cannot be explored in the current study which aim was to demonstrate the impact of birth weight. For example, inadequate diet or a lack of physical activity are known risk factors of dog overweight [1,72]. In addition, recent studies have demonstrated a strong association reported between a deletion present in the POMC gene of Labradors and increased body weight [73] and obesity [74]. Thus, future studies on genotype, including more dogs, are also warranted to evaluate at the same time numerous characteristics including genetics or diet for example. Life-long follow-up of a cohort of dogs would help to conduct this specific work.

## 5. Conclusions

To conclude, this work suggests that birth weight, interesting parameter to predict neonatal life, is also an important health indicator of long-term impact associated with adiposity at adulthood. Thus, it should be considered in overweight management plans: puppies born with low birth weight should be identified as being at higher risk of overweight and their owners should be encouraged to enter their dogs in specific weight control plans from an early stage in life. In the future, it may also be of interest to consider and evaluate the effect of early growth that may accentuate or temper the effect of birth weight on adult adiposity.

## Figures and Tables

**Figure 1 vetsci-10-00208-f001:**
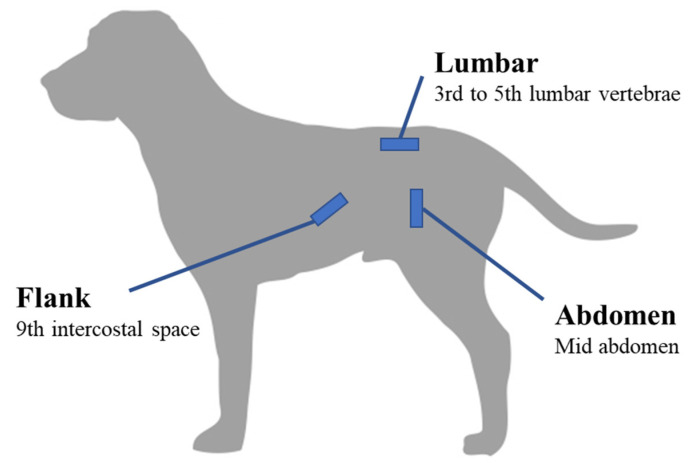
Schematic representation of the three anatomical sites used in the study to evaluate subcutaneous fat thickness in Labradors using real-time ultrasonography (*n* = 88, [32]).

**Figure 2 vetsci-10-00208-f002:**
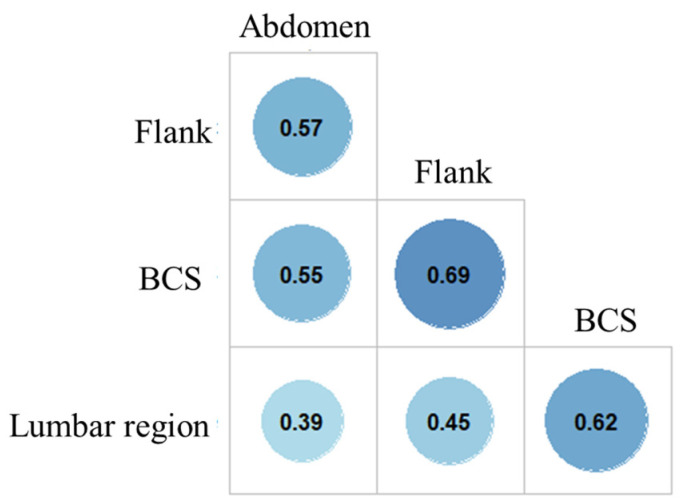
Correlation between body condition score and subcutaneous fat thickness assessed at three different locations (*n* = 88 Labradors). Note: BCS: body condition score; Pearson’s correlation coefficient is given for bivariate correlation between all pairs (*p* < 0.001 for all).

**Figure 3 vetsci-10-00208-f003:**
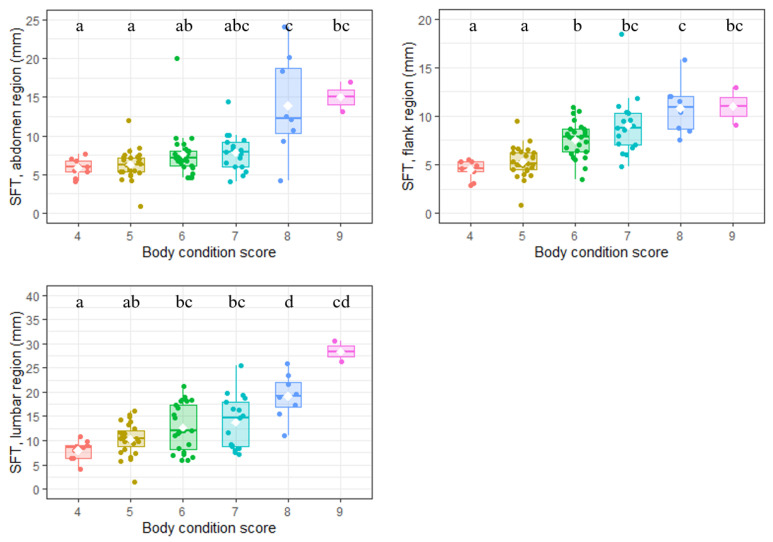
Relationship between body condition score and subcutaneous fat thickness (*n* = 88 Labradors). Note: SFT was measured at three anatomical sites (flank, abdomen and lumbar region). Within each box the different horizontal lines, from bottom to top, represent the 25th percentile, median and 75th percentile values. The white diamonds denote the mean value in the considered group. For each plot, groups sharing a letter were not significantly different (*p* > 0.05). Pairwise comparisons were conducted using a Dunn test with Benjamini–Hochberg correction following a Kruskal–Wallis rank sum test showing a significant relationship between BCS and SFT at the anatomical site considered. The scales of the y-axes are different between the three graphs.

**Figure 4 vetsci-10-00208-f004:**
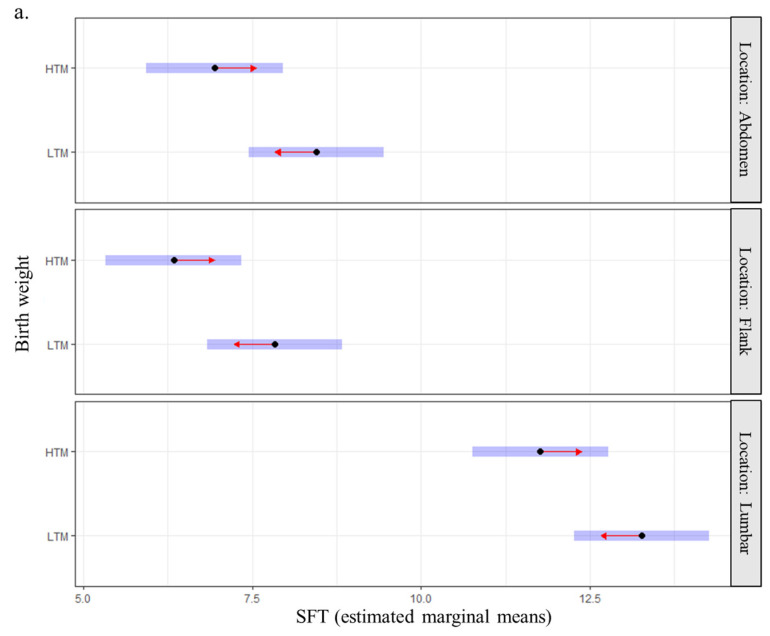
Relationship between birth weight and subcutaneous fat thickness at adulthood (*n* = 88 Labradors) (**a**) by anatomical site and (**b**) by age. Note: Blue bars indicate confidence intervals for estimated marginal means of SFT. Overlapping between red arrows from two groups indicates a non-significant difference between the two groups; HTM: higher than the median; LTM: lower than the median.

**Table 1 vetsci-10-00208-t001:** Description of adiposity parameters in study population (*n* = 88 Labradors).

Parameter	Mean	SD	Minimum	Maximum
BCS	5.9	1.2	4	9
SFT (all locations)	9.2	4.9	0.8	35.6
SFT Abdomen	7.8	3.7	0.9	24.1
SFT Flank	7.2	2.9	0.8	18.5
SFT Lumbar region	12.7	5.8	1.4	35.6

BCS: body condition score (1 = emaciated to 9 = grossly obese); SFT: subcutaneous fat thickness (in mm); SD: standard deviation.

**Table 2 vetsci-10-00208-t002:** Linear mixed regression model analysis for subcutaneous fat thickness (*n* = 88 Labradors).

Parameter		Coeff.	SE	*p*
	Intercept	6.954	0.785	
Age	Young, ≤2 years			<0.001
	Adult, 2–5 years	−0.462	0.754	
	Old, ≥5 years	3.061	0.720	
Neuter status	Entire			0.038
	Sterilized	1.251	0.603	
Birth weight	Lower than the median			0.014
	Higher or equal to the median	−1.500	0.613	
Location	Abdomen			<0.001
	Flank	−0.611	0.491	
	Lumbar	4.821	0.491	

Dog identity was included in the model as a random term. Coeff, coefficient; SE, standard error.

## Data Availability

Anonymized dataset is available from the Appendix A linked to this article.

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
