# Peer review of "Association between Birth Weight and Subcutaneous Fat Thickness at Adulthood in Dogs"

_vetsci, 2023, doi:10.3390/vetsci10030208_

Round 1

Reviewer 1 Report

Feedback to the authors regarding the manuscript “Association between low birthweight and subcutaneous fat thickness at adulthood in dogs” (Manuscript ID vetsci-2249480)

Brief summary 

The manuscript is well written in a fluent language with a novel content of scientific value. The authors have identified birthweight to be a risk factors of canine overweight in adulthood by evaluation of 88 Labrador retriever dogs. The manuscript is overall well-structured and the content is relevant for the field. The number of included dogs is ambitious and a relevant statistic model with correction for multiple comparisons and adjustments for animal characteristics has been applied.

I have made comments and suggestions on each section below, and I have also written some questions that you may try to answer and when relevant, also add the information to the manuscript text.

I look forward to read the manuscript again after revision.

Sincerely

General concept comments

The cited references could in some areas be improved (e.g. more updated references exist on overweight prevalence in dogs, please also add canine references on the reproducibility regarding BCS assessment), but the citation is otherwise ok.

The methodological descriptions could somewhat be improved. The anatomical locations of the regions that have been measured with RTU should for clarification preferably be described to the reader also in this current manuscript (not only in ref 26 as is now, as the use of the word “flanks” could be troublesome, see explanation below).

The naming “flanks” is somewhat troublesome as the exact anatomic location of the “flanks” (if I have understood it correctlyJ) is direct caudally of the last rib extending to direct cranially of pelvis (ala ossis ili). When reading ref 26 I understand that the region “flanks” in that validation study refers to the lateral location over the 9th intercostal space. “Abdomen” and “flank” with the descriptions currently used in this manuscript, could therefore be mistaken to be the same region!

Could you please describe the exact anatomical locations of your regions measured with RTU in your manuscript? For example:

“Laterally over the 9th intercostal space” (ribs, or flanks (but I personally prefer ribsJ))

“Over 3rd to 5th lumbar vertebrae” (lumbar)

“Mid lateral abdomen“ (abdomen)

Specific comments 

Abstract:

Line 26: Please define at which age the Labradors were evaluated, at the moment a definition of “adulthood” is lacking in the abstract and simple summary.

Line 28: “LTM / HTM for lower / higher or equal to the median” – could you please capitalise the letters that compose these definitions? I took me a while to understand those abbreviations.

Line 30: The following sentence is not clear to me: “The results showed that SFT values were higher in the lumbar region (compared to the other regions?), increased with age (at the lumbar region or all regions?), were higher in sterilised than in entire dogs (at the lumbar region or all regions?), and (something missing here in the sentence?) for LTM dogs than for HTM ones”. Please rewrite this sentence.

Introduction:

Line 39: Refs 1-2 are publications of great impact but they should be complemented with more recent references of prevalence of dog overweight, to be accurately used in this context.

Line 43: No original references of life expectancy, this should be added.

Line 46: Please clarify from which species the theory is generated. Are the references 8-11 based on human studies? (Please state species in the text for the reader)

Line 61: Is it true that BCS assessments lack repeatability in dogs? (inter- or intra repeatability). The references that you have put regards cows and sheep [24,25]. Please variegate this statement and include references on dogs.

Line 65: Please state that this reference (26) regards dogs.

Material and methods:

Line 79: Please clarify if the two assessors performed a joint assessment or if a mean was calculated from their respective assessment (I think you mean joint bit I am not sure J).

Line 83: Would it be possible to describe the exact anatomical location in this manuscript in addition to refer to the validated method (ref 26)? I think that would facilitate for the reader a lot. And please re-consider the use of the naming “flanks”.

Line 84: Were RTU measurements performed in standing position? Which level of restraint was needed? The study design is indeed non-invasive, but I miss a more clear method description (than just referring to ref 26). Please also add information on that Real-time utrasonography (RTU) was used on line 84, as well as information on, on which side of the animals that was evaluated witg RTU (sinister, dexter or random)

Results:

Line 118: The prevalence of overweight dogs in the cohort is quite high. It would be interesting for the reader to have the exact number of dogs within each score (in addition to mean, SD, median and rage). Also the number of dogs in each age category would be interesting to add.

Table 1: Pease add “in dogs” in the title (n=88 could otherwise be misunderstood as number of parameters). Unit for SFT is missing in the table, please add that to the table legend.

Figure 1: Pease add “in dogs” in the title (n=88 could otherwise be misunderstood as number of comparisons).

Figure 2: Please add “in dogs” in the title. Could you please describe what the boxplots represent (mean or median, percentiles etc). In the figure legends of both figure 1 and; could it be wise to remind the reader that you have analysed data with a linear mixed regression model with SFT as response variable? By just looking at the figures, the correlations/relationships might be a bit hard to interpret if you have not read the statistical part in detail and noted your advanced analytical model.

Line 146: Should age be presented before neutering? (as in appears in the table).

Figure 3: Please add “in dogs” in the title.

Discussion:

Line 171: Please add the reference you are comparing with.

Line 174: Could you please clarify what you refer to with “similar life guidelines at adulthood”?

Line 185: Would it be worth adding a citation of pomc-mutation in Labradors, predisposing the breed for adiposity?

Line 193: This slight increase of fat thickness (mm) is a bit hard for the reader to grasp at first sight when evaluating the figure as the numbering of the y-axis in figure 2 differ between a, b, and c. Please reconsider de numbering of axis for this reason, or consider highlight in the figure text the differences of axis.

Line 195: Maybe add “of fat deposits”?

Line 212: Might be?

Lines 216-219: Refs 56-58 need to have the species studied presented in the text (it is at the moment unclear fort the reader).

Line 224: Please add body fat distribution, “including visceral deposition” – at the moment you name both subcutaneous and subcutaneous plus visceral as “distribution”, which may be confusing to the quick reader.

Line 229: Maybe highlight the importance of the ability of body condition assessment of the owners? A control plan might include an educational plan for the owners, but that is not specified for the moment.

Lines 238-241: These lines have been added by mistake?

Line 255: Data Availability Statement is missing. Please add an adequate statement.

I look forward to read the manuscript again after revision!

Sincerely

Author Response

Responses to Reviewer 1’s Comments

Brief summary 

  • The manuscript is well written in a fluent language with a novel content of scientific value. The authors have identified birthweight to be a risk factors of canine overweight in adulthood by evaluation of 88 Labrador retriever dogs. The manuscript is overall well-structured and the content is relevant for the field. The number of included dogs is ambitious and a relevant statistic model with correction for multiple comparisons and adjustments for animal characteristics has been applied.

I have made comments and suggestions on each section below, and I have also written some questions that you may try to answer and when relevant, also add the information to the manuscript text.

            I look forward to read the manuscript again after revision.

Sincerely

We thank the reviewer for their kind words.

General concept comments

  • The cited references could in some areas be improved (e.g. more updated references exist on overweight prevalence in dogs, please also add canine references on the reproducibility regarding BCS assessment), but the citation is otherwise ok.

Thank you for this remark. We have added some references (see below). We hope that these additions will be sufficient.

  • The methodological descriptions could somewhat be improved. The anatomical locations of the regions that have been measured with RTU should for clarification preferably be described to the reader also in this current manuscript (not only in ref 26 as is now, as the use of the word “flanks” could be troublesome, see explanation below).

The naming “flanks” is somewhat troublesome as the exact anatomic location of the “flanks” (if I have understood it correctlyJ) is direct caudally of the last rib extending to direct cranially of pelvis (ala ossis ili). When reading ref 26 I understand that the region “flanks” in that validation study refers to the lateral location over the 9th intercostal space. “Abdomen” and “flank” with the descriptions currently used in this manuscript, could therefore be mistaken to be the same region!

Could you please describe the exact anatomical locations of your regions measured with RTU in your manuscript? For example:

“Laterally over the 9th intercostal space” (ribs, or flanks (but I personally prefer ribsJ))

“Over 3rd to 5th lumbar vertebrae” (lumbar)

“Mid lateral abdomen“ (abdomen)

Thank you for these suggestions which will indeed make it easier to understand the method used for this study directly. Details on anatomical sites were provided and a Figure was added to give a visual representation.

Line 91 “For evaluation on the abdomen, the probe was placed in vertical position on the left lateral wall of the abdomen, midway between the linea alba and the tip of the transverse process of the lumbar vertebra. For evaluation on flanks, the probe was placed transversely to the rib on the external oblique abdominal muscle on the left side of the dog, over the 9th inter-costal space and just above the costochondral junction. For evaluation at the lumbar region, the probe was positioned between the 3rd and 5th lumbar vertebrae, parallel to the spinous process, 2 to 3 cm to the left of the midline.”

Concerning the naming “flanks”, we have chosen to keep the same name as in the article to which we refer to avoid confusion.

Specific comments 

Abstract:

  • Line 26: Please define at which age the Labradors were evaluated, at the moment a definition of “adulthood” is lacking in the abstract and simple summary.

We added “> 1 year” in the abstract.

  • Line 28: “LTM / HTM for lower / higher or equal to the median” – could you please capitalise the letters that compose these definitions? I took me a while to understand those abbreviations.

This suggestion helped us to reformulate our abstract. We chose to remove the definition of the two birth weight groups in the abstract for ease of reading.

  • Line 30: The following sentence is not clear to me: “The results showed that SFT values were higher in the lumbar region (compared to the other regions?), increased with age (at the lumbar region or all regions?), were higher in sterilised than in entire dogs (at the lumbar region or all regions?), and (something missing here in the sentence?) for LTM dogs than for HTM ones”. Please rewrite this sentence.

The end of the abstract has been repeated. We hope that this makes it easier to understand.

Introduction:

  • Line 39: Refs 1-2 are publications of great impact but they should be complemented with more recent references of prevalence of dog overweight, to be accurately used in this context.

We added three more recent references (2010, 2019 and 2020).

  • Line 43: No original references of life expectancy, this should be added.

The review conducted by Laflamme (2012) reports “decreased insulin sensitivity in moderately overweight dogs was associated with a 15% decrease in median life span” but that was a review. We added two original studies conducted in humans and evaluating the decrease in life expectancy associated with overweight and obesity.

  • Line 46: Please clarify from which species the theory is generated. Are the references 8-11 based on human studies? (Please state species in the text for the reader)

Indeed, that was not clear. We added “in humans” at the end of the sentence (Line 50).

  • Line 61: Is it true that BCS assessments lack repeatability in dogs? (inter- or intra repeatability). The references that you have put regards cows and sheep [24,25]. Please variegate this statement and include references on dogs.

Thank you for this interesting remark. In fact, after reviewing the literature, studies in dogs do not show a lack of repeatability. Even if this point is underlined in other species, we have withdrawn this qualifier. In parallel we added two new references addressing the issues of subjectivity and experience of the evaluator in dogs.

  • Line 65: Please state that this reference (26) regards dogs.

The sentence was modified to add this information.

Line 65 “SFT, measured by ultrasonography at several anatomical sites is correlated with the proportion of body adipose tissue in canine species”.

Material and methods:

  • Line 79: Please clarify if the two assessors performed a joint assessment or if a mean was calculated from their respective assessment (I think you mean joint bit I am not sure J).

We added a sentence to clarify this point.

Line 84 “Any disagreement between the two operators was resolved by consensus”.

  • Line 83: Would it be possible to describe the exact anatomical location in this manuscript in addition to refer to the validated method (ref 26)? I think that would facilitate for the reader a lot. And please re-consider the use of the naming “flanks”.

Please see responses in section “General comments”

  • Line 84: Were RTU measurements performed in standing position? Which level of restraint was needed? The study design is indeed non-invasive, but I miss a more clear method description (than just referring to ref 26). Please also add information on that Real-time utrasonography (RTU) was used on line 84, as well as information on, on which side of the animals that was evaluated witg RTU (sinister, dexter or random)

This part of the MM was completed.

Concerning your question on the level of restraint. It was low because our study concerned dogs destined to become guide dogs and therefore with a good level of listening skills. To visualize in practice what it was like, here is a picture taken during the data collection. We did not consider it relevant to detail the restraint in the MM. We hope that these changes are acceptable.

Results:

  • Line 118: The prevalence of overweight dogs in the cohort is quite high. It would be interesting for the reader to have the exact number of dogs within each score (in addition to mean, SD, median and rage). Also the number of dogs in each age category would be interesting to add.

As suggested, we added the number of dogs within each BCS and in each age category in the description of the population.

Line 132 “ages ranged from 1.1 to 9.6 years (median: 3.4 years) with 28 young dogs (≤ 2 years), 29 adults (2-5 years) and 31 old dogs (≥ 5 years).”

Line 135 “BCS varied between 4 and 9 (n = 9, n = 27, n = 25, n = 17, n = 8 and n = 2, for BCS levels 4, 5, 6, 7, 8 and 9 respectively; Table 1)”

  • Table 1: Pease add “in dogs” in the title (n=88 could otherwise be misunderstood as number of parameters). Unit for SFT is missing in the table, please add that to the table legend.

Figure 1: Pease add “in dogs” in the title (n=88 could otherwise be misunderstood as number of comparisons).

Done.

  • Figure 2: Please add “in dogs” in the title. Could you please describe what the boxplots represent (mean or median, percentiles etc). In the figure legends of both figure 1 and; could it be wise to remind the reader that you have analysed data with a linear mixed regression model with SFT as response variable? By just looking at the figures, the correlations/relationships might be a bit hard to interpret if you have not read the statistical part in detail and noted your advanced analytical model.

Thank you for flagging that this aspect was unclear. The legend of this Figure was modified. We hope that these changes are acceptable.

Line 159 “SFT was measured at three anatomical sites (flank, abdomen and lumbar region). Within each box the different horizontal lines, from bottom to top, represent the 25th percentile, median and 75th percentile values. The white diamonds denote the mean value in the considered group. For each plot, groups sharing a letter were not significantly different (p > 0.05). Pairwise comparisons were conducted using a Dunn test with Benjamini-Hochberg correction following a Kruskal-Wallis rank sum test showing a significant relationship between BCS and SFT at the anatomical site considered.”

  • Line 146: Should age be presented before neutering? (as in appears in the table).

Done

  • Figure 3: Please add “in dogs” in the title.

Done

Discussion:

  • Line 171: Please add the reference you are comparing with.

Done

  • Line 174: Could you please clarify what you refer to with “similar life guidelines at adulthood”?

Thank you for this remark. On re-reading we realise that it was confusing. When they work as guide dogs, they are regularly monitored by the breeding and training center, which provides them with kibble and the quantity to be given, as well as various recommendations (adequate activity, antiparasitics...). However, our population also includes retired dogs who are less followed up even if they have recommendations on the nutritional level in particular. To simplify and avoid abusive interpretations, we preferred to remove this part of the sentence.

  • Line 185: Would it be worth adding a citation of pomc-mutation in Labradors, predisposing the breed for adiposity?

Many thanks for this suggestion. We have added this point at the end of the discussion. We hope that change is acceptable.

Line 275 “Many other factors suggested or demonstrated to have effects on adiposity in dogs cannot be explored in the current study which aim was to demonstrate the impact of birth weight. For example, inadequate diet or a lack of physical activity are known risk factors of dog overweight [1,72]. In addition, recent studies have demonstrated a strong association reported between a deletion present in the POMC gene of Labradors and increased body weight [73] and obesity [74]. Thus, future studies on genotype, including more dogs, are also warranted to evaluate in the same time numerous characteristics including genetics or diet for example. Life-long follow-up of a cohort of dogs would help to conduct this specific work.”

  • Line 193: This slight increase of fat thickness (mm) is a bit hard for the reader to grasp at first sight when evaluating the figure as the numbering of the y-axis in figure 2 differ between a, b, and c. Please reconsider de numbering of axis for this reason, or consider highlight in the figure text the differences of axis.

Thank you for flagging this point. We have chosen to keep the existing graduations to avoid overwriting the graphs for the abdomen and flanks. We have added a sentence at the end of the legend to raise this point. We hope that this simple addition is acceptable.

Line 165 “The scales of the y-axes are different between the three graphs”.

  • Line 195: Maybe add “of fat deposits”? Line 212: Might be?

Many thanks for these suggestions which improve clarity of this part.

  • Lines 216-219: Refs 56-58 need to have the species studied presented in the text (it is at the moment unclear fort the reader).

We have added “in humans”.

  • Line 224: Please add body fat distribution, “including visceral deposition” – at the moment you name both subcutaneous and subcutaneous plus visceral as “distribution”, which may be confusing to the quick reader.

Done

  • Line 229: Maybe highlight the importance of the ability of body condition assessment of the owners? A control plan might include an educational plan for the owners, but that is not specified for the moment.

Thank you for this suggestion. We have amended the paragraph to include this as a perspective.

Line 265 “In addition, it would be wise to consider training pet owners who currently tend to underestimate the adiposity of their animal [31,71]. This would allow them to detect overweight as early as possible, without waiting for a visit to the veterinarian.”

  • Lines 238-241: These lines have been added by mistake?

Thank you for flagging this point… I forgot to remove this part of the template. It’s done now.

  • Line 255: Data Availability Statement is missing. Please add an adequate statement.

Done. Sorry for the omission.

Reviewer 2 Report

This study aimed to investigate the role of birth weight in the development of obesity  in adulthood in Labradors, a dog breed chosen for its predisposition to being overweight. The subject is intriguing and current due to its growing diffusion also in pets. The authors applied a practical and non-invasive method, previously described and validated in dogs, to objectively assess body condition. Furthermore, the importance of the fetal and neonatal stages on future health conditions was highlighted by warning owners and breeders of the risk of obesity in lighter puppies. Useful clinical implications are recognized in this study. 

The main criticism concerns the cutoff (i.e. the median value of birth weight) established to divide dogs into LTM and HTM groups from which the risk of adult obesity was inferred. This threshold does not appear to be the most suitable for identifying low birth weight puppies. The median birth weight recorded in this caseload was 410 g (LL 119). Therefore the two groups should be: LTM < 410 g and HTM ≥ 410 g, it is correct? Please specify in the text. 

Furthermore, it is not clear how many dogs belonged to the LTM group and how many to the HTM one. Was birth weight also considered as a continuous variable? Another aspect to be clarified is how the newborns were identified in order to be able to trace with absolute certainty the birth weight of each adult dog. Were puppies belonging to the same litter, mainly the lightest ones, considered and compared to the littermates? In my opinion this comparison could provide interesting data.

Finally, since diet, physical activity and body weight of adult dogs appear to be involved in obesity tendency as are BCS and SFT, these aspects should be included in the discussion.

Specific comments

Results do not allow to state that “dogs with the lowest birth weights have a greater thickness of subcutaneous fat at adulthood than the others (LL 15 and LL 32)” but rather that puppies weighting less than the median had higher SFT as adults. I suggest remove ‘low’ from the title as well. 

Abstract 

LL 22-24 these three lines are the same in the  summary (LL 9-11)

LL 28 please write in full the acronyms LTM and HTM ( being the first time)

Materials and Methods

LL 74-76 dogs were born and raised in the same kennel and have always lived there, is that correct?

LL 81-84 how many operators performed the ultrasound examinations? Please provide some details (in addition to the bibliographic reference) on the ultrasound method. Synthetic and precise anatomical references can allow to standardize the procedure.

LL 99-100 as already pointed out, this cutoff used to group low birth weight puppies could represent a bias

Results

LL 119-120 please provide the number of dogs included in the LTM and HTM groups

LL 123-127 please can you provide a p-value for each correlation?

Discussion

Diet (quality and quantity) and physical activity are the main responsible for BCS and SFT in adult dogs. No mention on these aspects is made in discussion. Furthermore, since dogs came from  the same breeding kennel (similar bloodlines and genetics are assumed), body weight of adult dogs could also provide complementary information to BCS and SFT on obesity tendency.

The mention of visceral adipose tissue as a parameter for assessing overweight is an interesting point.

LL 171-174 as already reported, it is not clear whether the dogs were adopted or remained in the kennel after two months of life. Please specify what ‘similar life guidelines’ means.  Were dogs with different habits excluded? what habits? 

LL 211  I don’t understand ‘As the interaction between birth weight and location was not significant’. Moreover, was the BCS 9 associated with LTM?

Author Response

Responses to Reviewer 2’s Comments

Comments and Suggestions for Authors

  • This study aimed to investigate the role of birth weight in the development of obesity in adulthood in Labradors, a dog breed chosen for its predisposition to being overweight. The subject is intriguing and current due to its growing diffusion also in pets. The authors applied a practical and non-invasive method, previously described and validated in dogs, to objectively assess body condition. Furthermore, the importance of the fetal and neonatal stages on future health conditions was highlighted by warning owners and breeders of the risk of obesity in lighter puppies. Useful clinical implications are recognized in this study. 

We thank the reviewer for its kind words.

  • The main criticism concerns the cutoff (i.e. the median value of birth weight) established to divide dogs into LTM and HTM groups from which the risk of adult obesity was inferred. This threshold does not appear to be the most suitable for identifying low birth weight puppies.

Thank you for flagging this point. On re-reading we realize that it could be a little bit confusing. To improve this point, we have replaced “low birth weight” by “birth weight” in some parts of the discussion (lines 197 and 261). We also have amended a paragraph on the discussion to highlight this point. We hope that changes are acceptable.

Line 240 “Given the sample size and to be consistent with our previous study [23], the effect of birth weight was explored by separating the dogs into two groups relative to the median birth weight in the population. To further explore and refine this relationship between birth weight and adult adiposity, it may be of interest to study the effect of more extreme values of birth weight (e.g., first and fourth quartiles or first and tenth deciles).”

  • The median birth weight recorded in this caseload was 410 g (LL 119). Therefore, the two groups should be: LTM < 410 g and HTM ≥ 410 g, it is correct? Please specify in the text. 

  • Furthermore, it is not clear how many dogs belonged to the LTM group and how many to the HTM one. Was birth weight also considered as a continuous variable?

We have specified the number of dogs belonging to each category as well as the threshold value in the first part of the results.

Line 139 “The median birth weight was 410 g (interquartile range, IQR: 380-470) with. 43 dogs in the HTM group (birth weight ≥ 410 g) and 45 dogs in the LTM group (birth weight < 410 g).”

  • Another aspect to be clarified is how the newborns were identified in order to be able to trace with absolute certainty the birth weight of each adult dog. Were puppies belonging to the same litter, mainly the lightest ones, considered and compared to the littermates? In my opinion this comparison could provide interesting data.

As in many breeding kennels, newborns are identified at birth using collars with various colors or by colored woolen collars. Our study population was composed of adult dogs whose birth weights were later recovered from the breeding kennel (the same for all). It is a big breeding kennel and dogs of the study population did not come from the same litters. This could be an interesting new way to explore the issue: grouping by scope.

  • Finally, since diet, physical activity and body weight of adult dogs appear to be involved in obesity tendency as are BCS and SFT, these aspects should be included in the discussion.

Many thanks for this suggestion. A paragraph has been added to the discussion.

Line 275 “Many other factors suggested or demonstrated to have effects on adiposity in dogs cannot be explored in the current study which aim was to demonstrate the impact of birth weight. For example, inadequate diet or a lack of physical activity are known risk factors of dog overweight [1,72]. In addition, recent studies have demonstrated a strong association reported between a deletion present in the POMC gene of Labradors and increased body weight [73] and obesity [74]. Thus, future studies on genotype, including more dogs, are also warranted to evaluate in the same time numerous characteristics including genetics or diet for example. Life-long follow-up of a cohort of dogs would help to conduct this specific work.”

Specific comments

  • Results do not allow to state that “dogs with the lowest birth weights have a greater thickness of subcutaneous fat at adulthood than the others (LL 15 and LL 32)” but rather that puppies weighting less than the median had higher SFT as adults. I suggest remove ‘low’ from the title as well. 

Thank you for this interesting remark. As suggested, “low” was removed from the title.

Abstract 

  • LL 22-24 these three lines are the same in the summary (LL 9-11)

Sorry for the redundancy. The first sentence of the abstract has been reworded.

Line 22 “Overweight affects nearly 40% of dogs. The objective of this study was to explore the hypothesis of the Developmental Origins of Health and Disease through the association between birth weight and adiposity in adult dogs.”

  • LL 28 please write in full the acronyms LTM and HTM (being the first time)

Following comments from other reviewers, this part of the abstract has been reworded and abbreviations have been removed.

Materials and Methods

  • LL 74-76 dogs were born and raised in the same kennel and have always lived there, is that correct?

To clarify this point we have added a part of a sentence.

Line 77 “[…] and they are housed in volunteer families.”

  • LL 81-84 how many operators performed the ultrasound examinations? Please provide some details (in addition to the bibliographic reference) on the ultrasound method. Synthetic and precise anatomical references can allow to standardize the procedure.

Thank you for these suggestions which will indeed make it easier to understand the method used for this study directly. Details on anatomical sites were provided and a Figure was added to give a visual representation.

Line 93 “For evaluation on the abdomen, the probe was placed in vertical position on the left lateral wall of the abdomen, midway between the linea alba and the tip of the transverse process of the lumbar vertebra. For evaluation on flanks, the probe was placed transversely to the rib on the external oblique abdominal muscle on the left side of the dog, over the 9th inter-costal space and just above the costochondral junction. For evaluation at the lumbar region, the probe was positioned between the 3rd and 5th lumbar vertebrae, parallel to the spinous process, 2 to 3 cm to the left of the midline.”

  • LL 99-100 as already pointed out, this cutoff used to group low birth weight puppies could represent a bias

Thank you for flagging this point. As mentioned above to improve this point, we have replaced “low birth weight” by “birth weight” in some parts of the discussion. We also have amended a paragraph on the discussion to highlight this point. We hope that changes are acceptable.

Results

  • LL 119-120 please provide the number of dogs included in the LTM and HTM groups

See above please.

  • LL 123-127 please can you provide a p-value for each correlation?

Many thanks for this suggestion. We have completed the legend of this figure. We hope that change is acceptable.

Discussion

  • Diet (quality and quantity) and physical activity are the main responsible for BCS and SFT in adult dogs. No mention on these aspects is made in discussion. Furthermore, since dogs came from the same breeding kennel (similar bloodlines and genetics are assumed), body weight of adult dogs could also provide complementary information to BCS and SFT on obesity tendency.

We have added a paragraph at then end of the discussion as mentioned above.

  • The mention of visceral adipose tissue as a parameter for assessing overweight is an interesting point.

Thank you for this comment.

  • LL 171-174. as already reported, it is not clear whether the dogs were adopted or remained in the kennel after two months of life. Please specify what ‘similar life guidelines’ means.  Were dogs with different habits excluded? what habits? 

Thank you for flagging this point. On re-reading we realize that this point could be unclear. When dogs work as guide dogs, they are regularly monitored by the breeding and training center, which provides them with kibble and the quantity to be given, as well as various recommendations (adequate activity, antiparasitics...). However, our population also includes retired dogs who are less followed up even if they have recommendations on the nutritional level in particular. To simplify and avoid abusive interpretations, we preferred to remove this part of the sentence.

  • LL 211.  I don’t understand ‘As the interaction between birth weight and location was not significant’.

To clarify this part of the discussion, we have added a sentence at the end of the description of the results of the model. We hope that change is acceptable.

Line 184 “None of the interactions tested (birth weight and location or birth weight and age) were significant.”

  • Moreover, was the BCS 9 associated with LTM?

In the current study we have explored the association between birth weight and SFT. We have explored the association between BCS and birth weight in a previous work. We have observed significantly more overweight/obese Labradors when birth weights were lightest.

Reviewer 3 Report

The work could be interesting but some information needs to be supplemented.
The study was conducted on 88 adult Labrador retrievers, but what was the litter size of each of them? This is a very important parameter in relation to weight. Were they born by natural or caesarean section? Were they suckled by their mother? What was the size/weight of their mothers?
A gene responsible for obesity has long been known in the breed, were all the dogs free of it? Or was this factor not evaluated?

Author Response

Responses to Reviewer 3’s Comments

Comments and Suggestions for Authors

  • The work could be interesting but some information needs to be supplemented.
    The study was conducted on 88 adult Labrador retrievers, but what was the litter size of each of them? This is a very important parameter in relation to weight. Were they born by natural or caesarean section? Were they suckled by their mother? What was the size/weight of their mothers? A gene responsible for obesity has long been known in the breed, were all the dogs free of it? Or was this factor not evaluated?

Thank you for these questions which help to improve clarity of some parts of the paper.

Concerning the other neonatal characteristics, we have suggested the need for further studies. Litter size, type of delivery, timing for suckling are some important neonatal parameters which influence weight (at birth or later). However, in the current study our aim was not to explore variation factors of weight but to evaluate its relationship with later adiposity (through SFT).

Line 273 “In addition to birth weight, further studies are required to assess the influence of neonatal events and namely early growth on overweight risk at adulthood.”

Finally, we have added a paragraph at the end of the discussion to address other factors related to overweight including POMC mutation.

Line 275 “Many other factors suggested or demonstrated to have effects on adiposity in dogs cannot be explored in the current study which aim was to demonstrate the impact of birth weight. For example, inadequate diet or a lack of physical activity are known risk factors of dog overweight [1,72]. In addition, recent studies have demonstrated a strong association reported between a deletion present in the POMC gene of Labradors and increased body weight [73] and obesity [74]. Thus, future studies on genotype, including more dogs, are also warranted to evaluate in the same time numerous characteristics including genetics or diet for example. Life-long follow-up of a cohort of dogs would help to conduct this specific work.”

We hope that changes and our response are acceptable.

Round 2

Reviewer 1 Report

Many thanks for your revised manuscript!

You have addressed all my concerns and I only have a few minor remaining remarks:

1. Is there a "1" in the title? Looks strange :)

2. Do you have a permission from the original authors to use the dog picture in fig 1? If so, please state in figure legend. If not, could it be plagiarism??

3. Please add "Note:" at the start of the sentence in the added text in the figure legend.

4. In table 2 "Labradors" is missing in the parenthesis.

5. Please check the spelling of "fetal" on line 269 (spelling differs from abstract).

Sincerely,

Author Response

Thanks for your feedback.

  1. Not to my knowledge... I don't see it on the version I download.
  2. Thank you for this comment... I thought that quoting the article in question in the caption was enough but indeed after checking it is not enough. So I modified the figure by taking a free image. 
  3. That have been added.
  4. Thank you for your keen eye. The word Labrador has been added to the corresponding caption.
  5. The word "foetal" has been replaced by the word "fetal" throughout the manuscript

Sincerely

Reviewer 3 Report

What was requested has been supplemented

Author Response

Thank for your feeback.